# Locate, Crop and Segment: Efficient abdominal CT image segmentation on CPU

Yinyin Luo[†], Yue Liu[†], Wenbin Liu, Jingheng Dai, Xunliang Xiao, and
Gang Fang[*]

Institute of Computing Science and Technology, Guangzhou University, Guangzhou,
510006, China
[†]Co-first authors
[*]Corresponding author
gangf@gzhu.edu.cn

**Abstract.** Although current deep learning based models have achieved
tremendous successes in medical segmentation tasks, the deployment of
such models on CPU only devices is still challenging due to the substantial computational resources required for segmentation inference, especially for 3D medical images. Small sized models capable of efficient
inference have been proposed to mitigate the computational overheads,
however these small models usually largely sacrifice the segmentation
accuracy. In order to tackle the challenge in compliance with the requirements of MICCAI FLARE 2024 Challenge Task 2, i.e. deploying
advanced 3D abdominal CT segmentation models in non-GPU environments while maintaining high accuracy, we introduce a multi-scale knowledge distillation method to train a student model that maximally retains
the segmentation performance of the teacher model. In order to improve
the segmentation performance of tiny organs and Overcome the quality
issues of pseudo-labels themselves, We also design a weighted composite
loss function to train the model. Furthermore, for efficient segmentation
inference on CPU only devices, we introduce a liver-based Z-axis Region-of-Interest (RoI) localization strategy which effectively improve the segmentation efficiency. Experiments on the MICCAI FLARE 2024 datasets
have shown significant improvements in both segmentation accuracy and
efficiency. The proposed method achieves an average organ Dice Similarity Coefficient (DSC) of 88.70% and a Normalized Surface Dice (NSD)
of 94.29% on the public validation set. In the FLARE 2024 Task2 online
validation, the method achieved an average organ Dice Similarity Coefficient (DSC) of 88.47% and a Normalized Surface Dice (NSD) of 94.71%,
with an impressive average inference time of 12.33 seconds. The code is
available at https://github.com/lay-john/FLARE24-Task2.

**Keywords:** Semi-supervised · Deep learning · Organ segmentation ·
Knowledge distillation.

## 1   Introduction

The automatic and accurate semantic segmentation of medical images is a fundamental problem in medical image analysis, which serves as a crucial step in computer-aided diagnosis, surgical navigation, visual enhancement, radiotherapy, and biomarker measurement systems[27]. Segmentation accuracy is the most important factor to be considered when developing segmentation applications to aid medical image analysis, as it directly affects the patient diagnosis and the efficiency of clinical workflows. Based on this consensus, various methods, e.g. algorithm based methods and deep learning based methods, have been proposed to tackle this problem in recent years. For instance, U-Net[23] has shown superior segmentation accuracy by effectively utilizing skip connections to preserve spatial information. The DeepLab[2] series enhances multi-scale context and detail precision through the introduction of atrous convolution and Conditional Random Fields (CRFs). nnU-Net[12] offers an adaptive framework that optimizes performance for various datasets, showcasing remarkable versatility and effectiveness. Attention U-Net[22] leverages attention mechanisms to focus on crucial features, further improving segmentation accuracy. These methods have all achieved excellent segmentation results in different applications, advancing the development of medical image analysis. Among these methods, it is highly noted that nnU-Net has consistently demonstrated outstanding performance in different medical image segmentation tasks.

However, most current research works, including nnU-Net, neglect the importance of the computational efficiency of segmentation inference, i.e. the inference time and computational resource required, especially for 3D medical images. For an example, for a complete semantic segmentation of a 3D abdominal Computerized tomography (CT) image using nnU-Net, it would approximately take 60 seconds and 8G GPU memory, which is unbearable in real diagnosis practice. In fact, the memory consumption and GPU usage of these methods have led to a huge demand for computing resources, posing considerable challenges to the industrial deployment of the method. To mitigate the computational overheads, small sized models capable of efficient inference have been proposed, for an example EfficientNet[26], however previous researchers find that using smaller models often entails a compromise in performance[10]. Thus, this limitation prompts a critical challenge: how can we deploy state-of-the-art abdominal segmentation models in non-GPU environments without compromising segmentation accuracy.

In order to tackle the challenge in compliance with the requirements of MICCAI FLARE 2024 Challenge Task 2, i.e. deploying advanced 3D abdominal CT segmentation models in non-GPU environments while maintaining high accuracy, we introduce knowledge distillation and a pre-processing strategy in this paper. Specifically, in MICCAI FLARE 2024 Challenge Task 2, it is required to perform semantic segmentation of abdominal organ CT images using a CPU-based algorithm on a notebook computer with an 8GB memory limit. The task, which uses the same dataset as FLARE2022[15], involves segmenting 13 organs from CT images provided by over 20 medical groups, the organ labels are illustrated in Fig. 1. The dataset includes 2050 cases for model training, 250 cases

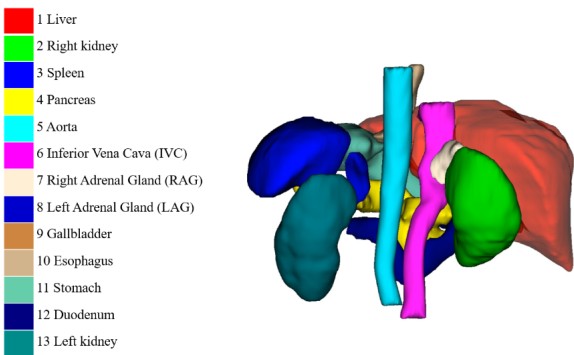

1 Liver
2 Right kidney
3 Spleen
4 Pancreas
5 Aorta
6 Inferior Vena Cava (IVC)
7 Right Adrenal Gland (RAG)
8 Left Adrenal Gland (LAG)
9 Gallbladder
10 Esophagus
11 Stomach
12 Duodenum
13 Left kidney

**Fig. 1.** Semantic labels of the 13 abdominal organs in FLARE 2024 Task2.

for validation, and 300 new, hidden test cases for final evaluation. The evaluation metrics include Dice Similarity Coefficient (DSC), Normalized Surface Dice (NSD), and seconds for inferring a single CT image(runtime).

Follow prior works, we also take advantage of small sized model for efficient inference, and we propose a multi-scale knowledge distillation method[31] to train a student model that maximally retains the segmentation performance of the teacher model. Knowledge distillation[9] is a technique where knowledge from a larger, more complex model (i.e. the teacher model) is transferred to a smaller, more efficient model (i.e. the student model), allowing the student model to achieve performance comparable to that of the teacher model while significantly reducing computational resources. Multi-scale knowledge distillation is commonly adopted in visual tasks to guide the training of a smaller student model by extracting multi-scale visual knowledge from a larger teacher model.Specifically, we perform multi-scale knowledge distillation by enforcing the student model to learn feature maps from the two intermediate layers of the teacher model by optimizing Mean Squared Error (MSE). In order to improve the segmentation performance of tiny organs and Overcome the quality issues of pseudo-labels themselves. we also propose a weighted composite loss algorithm based on DiceLoss and CELoss.

Furthermore, we introduce a pre-processing strategy to speed up the inference process. Given a 3D abdominal CT scan that is quite large in vertical size, the region of interest (RoI) for abdominal semantic segmentation may only occupy a portion of the scan, an example is shown in Fig. 2. It is unnecessary to perform inference on irrelevant areas. Thus, identifying and cropping the RoI is an effective strategy to reduce the inference time and the resource consumption. To address this, we propose a liver-based Z-axis RoI localization strategy to focus segmentation inference on the abdominal regions of interest. Previously, the approach for determining the RoI involved training a dedicated model specifically for RoI detection. This kind of methods induce increases in computational

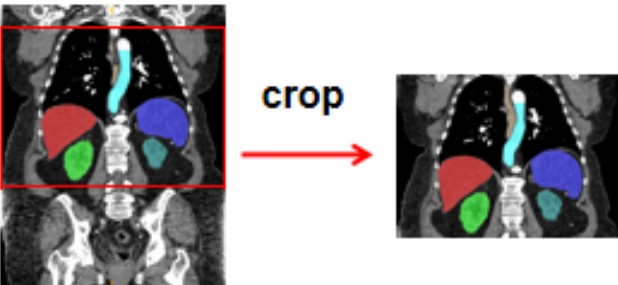

**Fig. 2.** The RoI only occupies a portion of the 3D abdominal CT scan.

complexity and time. Additionally, the RoI detection model often exhibits inferior performance compared to the segmentation model. Consequently, errors from the RoI detection model would propagate and adversely affect the final segmentation results. In contrast, our proposed strategy eliminates the need for a separate ROI detection model by leveraging our segmentation model to perform both ROI localization and segmentation. Our strategy is based on the accuracy of liver segmentation: as long as the Dice score for liver segmentation exceeds 60%, which is relatively easy to achieve, we can determine the ROI along the z-axis of the CT scan. This strategy minimizes the impact of poor input data quality and avoids the issues associated with a separate ROI detection model.

To summarize, in this paper, we propose a multi-scale knowledge distillation method and a Z-axis RoI localization strategy to tackle the MICCAR FLARE 2024 Challenge Task 2, the main contributions are summarized as follows: (1) Through multi-scale knowledge distillation, an efficient student model effectively absorbs the knowledge from the teacher model, thus maximally retains the performance of the teacher model while significantly reducing the computational overheads. (2) In order to improve the segmentation performance of tiny organs and Overcome the quality issues of pseudo-labels themselves, we also propose a weighted composite loss algorithm based on DiceLoss and CELoss. (3) To improve the speed of the segmentation inference, we further introduce a liver-based z-axis ROI localization strategy. Experimental results demonstrate the effectiveness of the proposed method in enhancing the performance of small sized model and accelerating the segmentation process.

## 2    Method

### 2.1    Preprocessing

- **Resample and normalization:**  We resample the pixel spacing to (2.2838, 1.8709, 1.8709) for all cases, and clip the pixel value based on the Hounsfield units to [-160, 240], and normalize all the cases in [0, 1] to ensure data stability and consistency.
- **Data augmentation:** In order to prevent the model from over-fitting, data augmentation is used in this study. The augmentation approaches of nnU-Net methodology have been utilized.

### 2.2    Network Architecture

Here we define two nnUNet structures of different model size: nnU-Net-Teacher and nnU-Net-Student. nnU-Net-Teacher is constructed as a conventional nnU-Net structure for medical image segmentation tasks. nnU-Net-Student is constructed by modifying a conventional nnU-Net architecture by changing the initial number of channels from 32 to 16 and adding an additional convolutional layer with a channel count doubled while keeping the resolution unchanged. The network architecture of nnU-Net-Teacher and nnU-Net-Student are shown in Fig. 3. The model hyper-parameters and the input patch size of [80, 160, 160] are chosen to satisfy the memory requirement by the FLARE 2024 Task2 competition.

### 2.3    Proposed Method

Specifically, this paper proposes a multi-scale knowledge distillation method that initially trains a large Teacher model using the provided FLARE22 dataset and pseudo labels. This Teacher model is subsequently utilized to perform multi-scale knowledge distillation on a smaller Student model. A distillation loss (MSE) is calculated at the lowest two layers of the encoders in both the Teacher model and the Student model, and this distillation loss is incorporated into the Student loss to enhance the performance of the Student model through multi-scale knowledge distillation.When training the model. we also proposed a weighted compound loss algorithm based on DiceLoss and CELoss.Another contribution is the liver-based z-axis ROI localization technique. This technique involves scanning from the bottom to the top along the z-axis and stopping when a certain number of liver voxels are detected, using this patch to determine the z-axis ROI.

  **Multi-scale Knowledge Distillation:** Initially, a large nnUNet model with an initial channel number of 32 is trained using Dice and Cross-Entropy loss for segmentation, achieving satisfactory performance. Subsequently, a smaller model is designed. To ensure consistency in the region where knowledge distillation is performed, an additional layer which does not change the resolution is added to the Student model, which has an initial channel number of 16, as illustrated in Fig. 3. During the training of the Student model, distillation losses are introduced

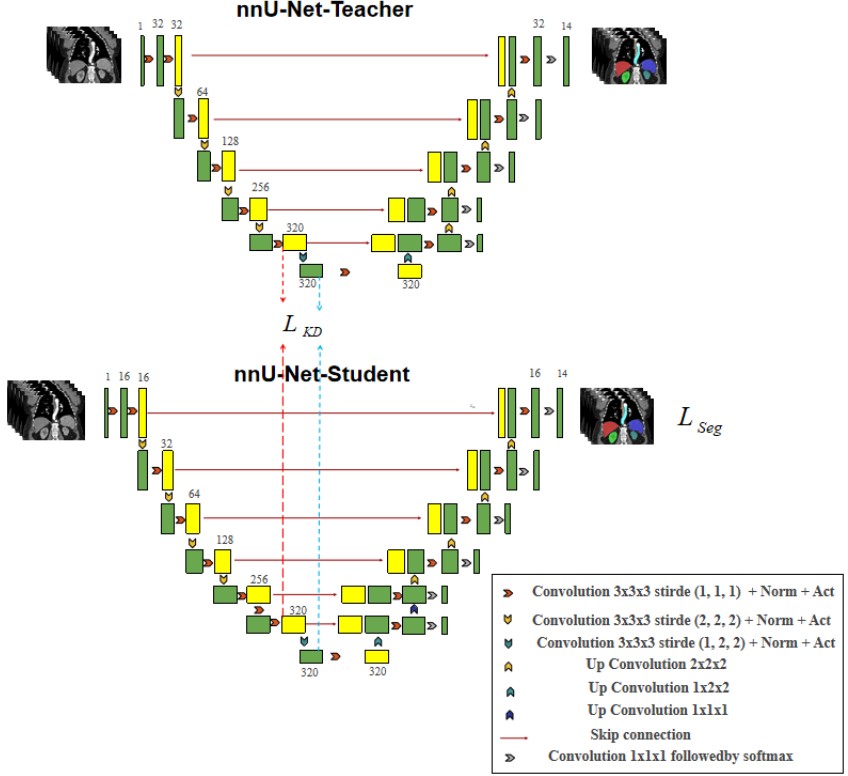

**Fig. 3.** Overview of the proposed method. (see Sect. 2.3 for details).

At the bottom two layers of the encoder. Both scales utilize MSE loss, and the distillation losses from these two scales are simply summed to obtain the total distillation loss. This total distillation loss is then multiplied by a weight w and added to the final loss.Here w is set to 0.5.Teacher model and Student model both use a deep supervision strategy.During the training phases of the two models, we use pseudo-labels generated from 2,000 unlabeled cases based on the winning solution[11] from FLARE 2022 , without any further processing.

**Loss Function:** The loss function combines weighted DiceLoss and CELoss, where CombLoss[25] converges significantly faster than cross-entropy loss. Specifically, different weights are applied to DiceLoss and CELoss. For the weight w1 of DiceLoss, we use the Mean DSC for each organ as reported in the winning solution of FLARE 2022[11]. For the weight w2 of CELoss, we allocate weights based on the size ratio of each organ, with smaller organs receiving higher weights. This approach ensures that the model focuses more on the quality of pseudo-labels and effectively considers the impact of organ size on the model.When train the Student model we need to add the total distillation loss.The loss function im-

proves the performance of the model. See Section 4.1 for details. The formula is as follows:

$$L_{Dice}(y, \hat{y}, w1) = \sum_{i}^{c_0} w1_i \left( 1 - \frac{2 \sum_{j=1}^{N} y_j^i \hat{y}_j^i}{\sum_{j=1}^{N} y_j^i + \hat{y}_j^i} \right) \tag{1}$$

$$L_{CE}(y, \hat{y}, w2) = \sum_{i}^{c_0} w2_i \left( -\frac{1}{N} \sum_{j=1}^{N} \left( y_j^i \log(\hat{y_j^i}) + (1 - y_j^i) \log(1 - \hat{y_j^i}) \right) \right) \tag{2}$$

$$L_{KD} = w3 \cdot MSE_1 + w4 \cdot MSE_2 \tag{3}$$

$$L_{Seg} = \alpha_{dc} \cdot L_{Dice}(y, \hat{y}, w1) + \alpha_{ce} \cdot L_{CE}(y, \hat{y}, w2) + \alpha_{KD} \cdot L_{KD} \tag{4}$$

Where the $y$ and $\hat{y}$ mean the ground truth and the predicted probability, respectively, and $N$ is the number of pixels. $\alpha_{dc}$ and $\alpha_{ce}$ are the hyperparameters to balance the contribution of DiceLoss and CELoss. $\alpha_{dc}$ and $\alpha_{ce}$ are set to 0.5 in this study. $\alpha_{KD}$ is the hyperparameter of knowledge distillation. In this study, it is set to 0.5. $w3$ and $w4$ are hyperparameters on two scales of multi-scale knowledge distillation, and they are both set to 0.5 in this study.

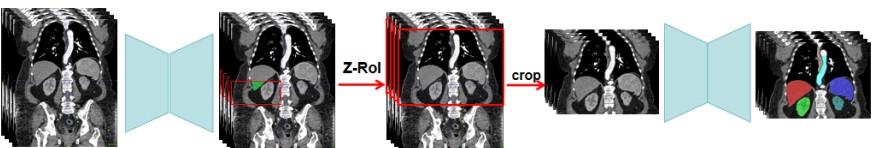

**Fig. 4.** The proposed liver-based Z-axis RoI localization strategy. (see Sect. 2.3 for details.)

**Liver-based Z-axis RoI Localization:** Prior to inference, the z-axis ROI is determined and the region is cropped for final inference, as shown in Fig. 4, we also wrote pseudocode for this process, as shown in Algorithm 1. Specifically, fixed sliding window regions are set for the x-axis and y-axis, both at $(0, 160)$, with a step size of 80 for sliding only along the z-axis starting from 0. A liver threshold, set to 20,000 in this study, is defined. During the z-axis sliding process, if the number of liver label voxels in the window is less than the threshold, the window slides to the next position; otherwise, the sliding stops. At this point,

the area of the window on the z-axis is $(i, i + \text{patch\_size}[0])$, where $i$ is the value of the lowest point of the z-axis window. Within this sliding window, the bounding box of the liver label is extracted, and based on the lower limit of the z-axis of this bounding box (minzidx), the final z-axis ROI range is $(i + \text{minzidx} - \frac{7}{8} \times \text{patch\_size}[0],\ i + \text{minzidx} + 2 \times \frac{7}{8} \times \text{patch\_size}[0])$. This provides a z-axis ROI region, which is then used for inference. The z-axis ROI region is defined as covering three z-axis sliding regions because with our patch_size = $(80, 160, 160)$, spacing = $(2.2838, 1.8709, 1.8709)$, and stride = $\left(\frac{7}{8}, \frac{7}{8}, \frac{7}{8}\right)$, all z-axis abdominal organ regions can be covered within three sliding windows.

---

**Algorithm 1** Pseudocode for axis-based z-axis RoI positioning.

---

**Input:** CT scan, liver threshold (20000), patch size (80, 160, 160), spacing (2.2838, 1.8709, 1.8709), stride (7/8, 7/8, 7/8)

**Output:** z-axis ROI region

1: **Initialize:** $x\_range$, $y\_range$, $z\_start$, $step\_size$, $liver\_threshold$, $patch\_size\_z$
2: **Define** sliding window regions for $x$ and $y$ axes:
3: $x\_window = x\_range$
4: $y\_window = y\_range$
5: **Slide** along the $z$-axis:
6: **for** $i = z\_start$ **to** $end\_of\_z\_axis$ **step** $step\_size$ **do**
7: $\quad current\_window = CT\_scan[x\_window, y\_window, i : i + patch\_size\_z]$
8: $\quad liver\_voxel\_count = \text{count\_liver\_voxels}(current\_window)$
9: $\quad$ **if** $liver\_voxel\_count \geq liver\_threshold$ **then**
10: $\quad\quad$ **break**
11: $\quad$ **end if**
12: **end for**
13: **Determine** $z$-axis ROI range:
14: $z\_lower\_bound = i$
15: **Extract** liver label bounding box within the sliding window:
16: $liver\_bbox = \text{extract\_liver\_bbox}(current\_window)$
17: $minzidx = liver\_bbox.z\_lower\_limit$
18: **Calculate** final $z$-axis ROI range:
19: $final\_z\_lower = z\_lower\_bound + minzidx - (7/8 \times patch\_size\_z)$
20: $final\_z\_upper = z\_lower\_bound + minzidx + (2 \times 7/8 \times patch\_size\_z)$
21: $z\_axis\_ROI = (final\_z\_lower, final\_z\_upper)$
22: **return** $z\_axis\_ROI$

---

### 2.4   Post-processing

In the post-processing stage, to save time, no post-processing operations were performed.

## 3   Experiments

### 3.1   Dataset and evaluation measures

The dataset is curated from more than 40 medical centers under the license permission, including TCIA [3], LiTS [1], MSD [24], KiTS [7,8], autoPET [6,5], AMOS [14], AbdomenCT-1K [21], TotalSegmentator [28], and past FLARE challenges [18,19,20]. The training set includes 2050 abdomen CT scans where 50 CT scans with complete labels and 2000 CT scans without labels. The validation and testing sets include 250 and 300 CT scans, respectively. The annotation process used ITK-SNAP [30], nnU-Net [13], MedSAM [16], and Slicer Plugins [4,17].

The evaluation metrics encompass two accuracy measures—Dice Similarity Coefficient (DSC) and Normalized Surface Dice (NSD)—alongside one efficiency measures—runtime. These metrics collectively contribute to the ranking computation. During inference, GPU is not available where the algorithm can only rely on CPU.

### 3.2   Implementation details

**Environment settings**  The development environments and requirements are presented in Table 1.

**Table 1.** Development environments and requirements.

| System | Ubuntu 22.04 LTS or Windows 10 |
|---|---|
| CPU | Intel(R) Core(TM) i9-10900X CPU@3.70GHz |
| RAM | 16×4×32GB; 2933MT/s |
| Programming language | Python 3.9.16 |
| Deep learning framework | torch 2.1.0, torchvision 0.16.0 |
| Specific dependencies | nnU-Net 1.7.0 |
| Code | https://github.com/lay-john/FLARE24-Task2 |

**Training protocols**  During the training phase, we set the batch size to 2 and randomly select all samples within each epoch. For each sample, we perform random patch cropping with patch sizes of (80, 160, 160). As for the optimizer, we utilize AdamW with a learning rate of 1e-2 and a weight decay of 1e-5. The learning rate updating follows the default mechanism of AdamW. Additional details are presented in Table 2.

**Table 2.** Training protocols for Teacher model.

| | |
|---|---|
| Network initialization | |
| Batch size | 2 |
| Patch size | $80 \times 160 \times 160$ |
| Total epochs | 500 |
| Optimizer | AdamW with weight decay($\mu$ = 1e -5) |
| Initial learning rate (lr) | 0.01 |
| Lr decay schedule | halved by 200 epochs |
| Training time | 35 hours |
| Loss function | DiceLoss and CELoss |
| Number of Teacher model parameters | 30.8M[1] |
| Number of Teacher model flops | 469.2262 G[2] |
| Teacher model $CO_2$eq | 1.61908 Kg[3] |
| Number of Student model parameters | 30.3M[4] |
| Number of Student model flops | 128.1024 G[5] |
| Student model $CO_2$eq | 0.73306 Kg[6] |

## 4    Results and discussion

### 4.1    Quantitative results on validation set

Quantitative evaluation results are shown in Table 3, demonstrating that the proposed method achieves very promising results for major organs such as the liver, spleen, kidneys, and stomach. However, segmenting smaller organs remains highly challenging and requires further attention, particularly for very small organs with unclear boundaries, such as the adrenal glands and duodenum.

To conduct a more comprehensive ablation study of our proposed method, we performed quantitative experiments, as shown in Table 4. Both of our proposed methods have shown improvements.

**Table 3.** Quantitative evaluation results.

| Target | Public Validation | | Online Validation | |
|---|---|---|---|---|
| | DSC(%) | NSD(%) | DSC(%) | NSD(%) |
| Liver | 97.09 | 98.41 | 97.14 | 98.12 |
| Right Kidney | 92.37 | 94.26 | 93.87 | 95.71 |
| Spleen | 96.82 | 98.89 | 96.11 | 98.07 |
| Pancreas | 89.25 | 97.80 | 86.13 | 96.50 |
| Aorta | 94.50 | 98.48 | 94.78 | 98.85 |
| Inferior vena cava | 89.27 | 91.02 | 89.48 | 92.20 |
| Right adrenal gland | 79.58 | 93.01 | 81.71 | 95.63 |
| Left adrenal gland | 79.64 | 91.86 | 81.48 | 94.57 |
| Gallbladder | 84.58 | 86.06 | 83.03 | 84.86 |
| Esophagus | 85.77 | 95.54 | 81.25 | 92.35 |
| Stomach | 91.38 | 94.41 | 91.96 | 95.92 |
| Duodenum | 81.68 | 94.41 | 79.65 | 92.79 |
| Left kidney | 91.18 | 92.66 | 93.53 | 95.72 |
| Average | 88.70 | 94.29 | 88.47 | 94.71 |

**Table 4.** Overview of Ablation Experiment Results. Proposed: Base+KD+Z-RoI.

| Target | Base | | Base+ KD | | Base+KD+Z-RoI | |
|---|---|---|---|---|---|---|
| | DSC(%) | NSD(%) | DSC(%) | NSD(%) | DSC(%) | NSD (%) |
| Liver | 96.59 | 97.97 | 97.07 | 98.25 | 97.09 | 98.41 |
| Right Kidney | 89.97 | 91.10 | 90.92 | 92.57 | 92.37 | 94.26 |
| Spleen | 96.68 | 98.78 | 96.81 | 98.86 | 96.82 | 98.89 |
| Pancreas | 88.07 | 97.41 | 88.91 | 97.43 | 89.25 | 97.80 |
| Aorta | 93.84 | 98.13 | 94.56 | 98.60 | 94.50 | 98.48 |
| Inferior vena cava | 85.93 | 87.43 | 88.83 | 90.58 | 89.27 | 91.02 |
| Right adrenal gland | 76.69 | 90.65 | 79.79 | 93.22 | 79.58 | 93.01 |
| Left adrenal gland | 74.77 | 87.59 | 76.05 | 88.34 | 79.64 | 91.86 |
| Gallbladder | 77.45 | 79.64 | 84.96 | 86.70 | 84.58 | 86.06 |
| Esophagus | 82.58 | 92.25 | 84.60 | 94.27 | 85.77 | 95.54 |
| Stomach | 88.93 | 92.87 | 91.12 | 94.75 | 91.38 | 94.41 |
| Duodenum | 80.26 | 92.10 | 81.09 | 93.66 | 81.68 | 94.41 |
| Left kidney | 86.89 | 89.45 | 88.01 | 89.60 | 91.18 | 92.66 |
| Average | 86.05 | 91.95 | 87.90 | 93.60 | 88.70 | 94.29 |

To visually demonstrate the impact of our method on inference speed on cpu, we conducted quantitative experiments on inference speed, as shown in Table 5. the length of the step is [7/8, 7/8, 7/8] times the window width for each axis.

## 4.2    Qualitative results on validation set

In this section, we show the two good segmentation cases and two bad segmentation case, along with the time consumption for inference on several large CT scans.

**Table 5.** Overview of Ablation Experiment Results on inference speed. Proposed: Small nnU-Net + Z-RoI. Time is measured in seconds.

| Target | Average time(s) |
|---|---|
| Big nnU-Net | 38.21 |
| Small nnU-Net | 17.88 |
| Small nnU-Net + Z-RoI | 12.33 |

**Good segmentation cases:** Fig. 5 presents examples of good segmentation results. In FLARETs_0001, both the Baseline and our proposed method demonstrate strong segmentation performance. In FLARETs_0012, while the overall segmentation performance remains satisfactory, our method exhibits a minor shortcoming where a portion of the right kidney is erroneously excluded and classified as background. This discrepancy may stem from our model's incorrect learning from the teacher model, leading to reduced sensitivity in certain localized regions, thereby failing to capture this specific area.

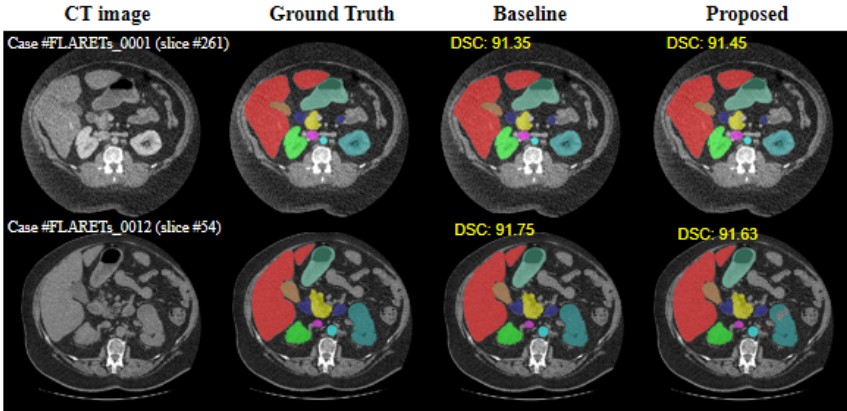

**Fig. 5.** Good segmentation cases from 50 validation set.

**Bad segmentation cases:** Fig. 6 presents examples of bad segmentation results.In FLARETs_0033, the baseline model demonstrates suboptimal performance, occasionally segmenting a single organ as two distinct organs. In contrast, our proposed method generally performs well; however, it misclassifies a portion of the duodenum as the stomach and exhibits poor segmentation performance for the pancreas. This is likely due to the anatomical proximity of the stomach and duodenum, especially at the junction between the lower end of the stomach and the beginning of the duodenum, where the boundary between the two organs is not always clearly defined. Additionally, the segmentation performance

of the pancreas was suboptimal. This may because the pancreas is located deep within the abdomen, surrounded by other organs such as the stomach, duodenum, and liver, and has a long and variable shape. These anatomical features make it difficult to distinguish the pancreas from other tissues in CT images. In FLARETs_0049, Both Baseline and our method have poor segmentation of the liver and inferior vena cava. As can be seen from the figure, there are patches of black areas in the liver of the FLARETs_0049 use case. These may be some diseased areas, resulting in poor model performance. For the inferior vena cava, the anatomical location of the inferior vena cava is close to many other important organs and blood vessels. The morphology may not be significantly different from the surrounding tissue, making it difficult for the model to accurately distinguish the inferior vena cava from the inferior vena cava and the shape and size of the inferior vena cava may vary greatly between different patients. This variation may increase the difficulty of model segmentation, which is difficult for our model to learn.

Upon further investigation, we recognize that the limitations of our model extend beyond the anatomical complexities. Specifically, the feature learning capabilities of our current model architecture may be insufficient for distinguishing the inferior vena cava from surrounding tissues. The convolutional layers in our network might struggle to capture fine-grained anatomical details, especially in regions with high variability. Additionally, the current loss function design may not adequately penalize errors in regions with high anatomical variability, such as the inferior vena cava. This suggests that the model may benefit from alternative loss functions that incorporate anatomical priors or use a combination of different loss terms to improve segmentation accuracy.

To address these limitations, we propose several targeted improvement plans. First, we will experiment with different network architectures, such as those incorporating attention mechanisms or multi-scale feature fusion, which are better suited for handling complex anatomical structures. Second, we will investigate advanced training strategies, including curriculum learning and data augmentation techniques tailored to the specific challenges of the inferior vena cava. These improvements aim to enhance the model's ability to learn from the data and improve segmentation performance in challenging anatomical regions.

Table 6 presents the time taken for several large CT scans, indicating that our method has a significant advantage in reducing inference time, especially for large CT scans.

**Table 6.** Overview of Ablation Experiment Results on inference speed. Proposed: Small nnU-Net + Z-RoI.

| Dataset | Big nnU-Net(s) | Small nnU-Net(s) | Small nnU-Net + Z-RoI(s) |
| --- | --- | --- | --- |
| FLARETs_0001_0000 | 62.79 | 29.29 | 15.72 |
| FLARETs_0010_0000 | 62.48 | 29.56 | 17.05 |
| FLARETs_0050_0000 | 114.84 | 55.58 | 25.39 |

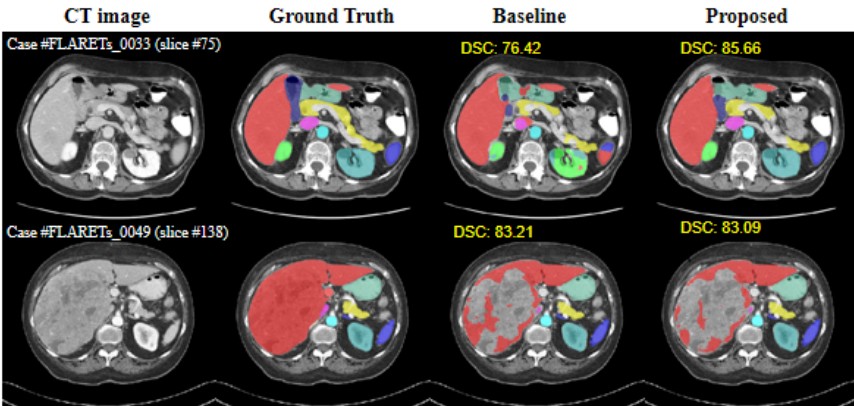

**Fig. 6.** Bad segmentation cases from 50 validation set.

### 4.3   Segmentation efficiency results on validation set

We quantitatively evaluate the segmentation efficiency of our model based on running time, as shown in Table 7.

**Table 7.** Quantitative evaluation of segmentation efficiency in terms of the running time. Evaluation CPU: Intel(R) Core(TM) i5-12400F CPU @ 2.50GHz × 12.

| Case ID | Image Size | Running Time (s) |
|---------|------------|------------------|
| 0059 | (512, 512, 55) | 10.66 |
| 0005 | (512, 512, 124) | 6.62 |
| 0159 | (512, 512, 152) | 11.20 |
| 0176 | (512, 512, 218) | 9.68 |
| 0112 | (512, 512, 299) | 10.36 |
| 0135 | (512, 512, 316) | 10.53 |
| 0150 | (512, 512, 457) | 7.94 |
| 0134 | (512, 512, 597) | 12.77 |

### 4.4   Results on final testing set

Tables 8,  9 and  10 present the final testing results of our proposed method in the FLARE 2024 Task2 across the Asian, European, and North American datasets, respectively. These tables list the performance metrics of our method, including the Dice Similarity Coefficient (DSC), Normalized Surface Distance

(NSD), and inference time. Each metric is reported with both the mean and standard deviation (Mean $\pm$ Std), as well as the median along with the first and third quartiles (Median (Q1, Q3)).

**Table 8.** Final testing results of the proposed method on the FLARE 2024 Task2 Asian datasets.

| Metric | Mean $\pm$ Std | Median (Q1, Q3) |
|---|---|---|
| DSC (%) | 85.2 $\pm$ 6.2 | 87.8 (80.7, 90.5) |
| NSD (%) | 92.1 $\pm$ 5.9 | 94.3 (88.3, 96.8) |
| Inference Time (s) | 15.5 $\pm$ 3.7 | 13.7 (12.8, 18) |

**Table 9.** Final testing results of the proposed method on the FLARE 2024 Task2 European datasets.

| Metric | Mean $\pm$ Std | Median (Q1, Q3) |
|---|---|---|
| DSC (%) | 87.4 $\pm$ 6.2 | 89.7 (84.7, 91.9) |
| NSD (%) | 93.4 $\pm$ 6.1 | 95.7 (91.3, 98) |
| Inference Time (s) | 16.4 $\pm$ 4.6 | 17.5 (12.8, 18.3) |

**Table 10.** Final testing results of the proposed method on the FLARE 2024 Task2 North American datasets.

| Metric | Mean $\pm$ Std | Median (Q1, Q3) |
|---|---|---|
| DSC (%) | 87.6 $\pm$ 4.5 | 89.1 (85.4, 91) |
| NSD (%) | 93.1 $\pm$ 4.7 | 94.7 (91, 96.7) |
| Inference Time (s) | 12.6 $\pm$ 2.7 | 13 (12.7, 13.3) |

### 4.5   Limitation and future work

In this study, the segmentation performance for small organs remains unsatisfactory, especially the boundary segmentation of these organs is not very clear. In future work, we will focus on improving the segmentation of these small organs and improve the boundary segmentation effect of these small organs, such as the gallbladder and adrenal glands. Additionally, we only determined the roi of the z-axis and did not consider the roi positioning of the x-axis and y-axis. In future work, we will explore the positioning scheme of the x-axis and y-axis to achieve the positioning scheme of the three axes.

## 5    Conclusion

To facilitate efficient semantic segmentation inference of abdominal CT scans on a CPU, this paper introduces a multi-scale knowledge distillation method to improve the performance of small models and design a weighted Composite loss function to alleviate class imbalance problem and Overcome the quality issues of pseudo-labels themselves. At inference, we further introduce a liver-based z-axis ROI localization strategy to accelerate inference. Quantitative and qualitative results demonstrate that our method can efficiently and flexibly learn information about multiple organs from the dataset. We validated our method on the MICCAI FLARE 2024 challenge dataset, proving that the proposed approach performs excellently in segmenting 13 different organs on a CPU.

**Acknowledgements** The authors of this paper declare that the segmentation method they implemented for participation in the FLARE 2024 challenge has not used any pre-trained models nor additional datasets other than those provided by the organizers. The proposed solution is fully automatic without any manual intervention. We thank all data owners for making the CT scans publicly available and CodaBench [29] for hosting the challenge platform.

## Disclosure of Interests

The authors declare no competing interests.

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

**Table 11.** Checklist Table. Please fill out this checklist table in the answer column.

| Requirements | Answer |
|---|---|
| A meaningful title | Yes |
| The number of authors (≤6) 6 | Number |
| Author affiliations and ORCID | Yes |
| Corresponding author email is presented | Yes |
| Validation scores are presented in the abstract | Yes |
| Introduction includes at least three parts: background, related work, and motivation | Yes |
| A pipeline/network figure is provided | 6 |
| Pre-processing | 5 |
| Strategies to improve model inference | 5-8 |
| Post-processing | 8 |
| The dataset and evaluation metric section are presented | 9 |
| Environment setting table is provided | 1 |
| Training protocol table is provided | 2 |
| Ablation study | 10-11 |
| Efficiency evaluation results are provided | 3 |
| Visualized segmentation example is provided | 5 6 |
| Limitation and future work are presented | Yes |
| Reference format is consistent. | Yes |