# OpenReview forum: "Locate, Crop and Segment: Efficient abdominal CT image segmentation on CPU"
_MICCAI.org/2024/Challenge/FLARE — FLARE 2024 withMinorRevisions_

### Official Review · Reviewer_HZT7 · 2025-01-20
**Review of "Locate, Crop and Segment: Efficient abdominal CT image segmentation on CPU"**

**Rating:** 9
**Confidence:** 5

**Review:**

This is a very complete paper. This paper presents a CPU-based approach for efficient abdominal multi-organ segmentation using knowledge distillation and ROI localization strategies. The proposed liver-based ROI localization strategy is particularly novel, providing an efficient way to identify regions of interest without requiring additional detection models.
Minor revision suggested:

The formulas in the section "Liver-based Z-axis RoI Localization" should be written in equation format rather than text format. For example:

Replace "i + minzidx - 7/8 * patch_size[0]" with proper mathematical notation
Format the equations for ROI boundary calculations using standard mathematical typography

---

> ### Author Response · Authors · 2025-03-29
> **Response to Reviewer HZT7**
>
> Thank you for your comment. We have already replaced part of the formula with the appropriate mathematical notation in the article.

---

### Official Review · Reviewer_MzYT · 2025-01-27
**Review of "Locate, Crop and Segment: Efficient abdominal CT image segmentation on CPU"**

**Rating:** 9
**Confidence:** 4

**Review:**

This article aims to address the issue of efficient abdominal CT image segmentation on CPU devices. The study targets Task 2 of the MICCAI FLARE 2024 challenge, introducing a multi-scale knowledge distillation method to train the student model to retain the segmentation performance of the teacher model, designing a weighted composite loss function to improve the segmentation of small organs and handle the quality of pseudo-labels, and proposing a Z-axis ROI localization strategy based on the liver to accelerate inference. Through experiments on relevant datasets, the method has achieved significant results in both segmentation accuracy and efficiency, with outstanding performance on both the public validation set and the online validation set, providing an effective solution for abdominal CT image segmentation in non-GPU environments.
Here are some small suggestions:
1. Small organ segmentation effect issue: Although the overall segmentation performance is good, there are still challenges in segmenting small organs such as adrenal glands and duodenum, with unclear boundary segmentation, which may affect the accurate diagnosis of related diseases.
2.  ROI localization limitation issue: Only the Z-axis ROI is determined, without considering the ROI localization of the X-axis and Y-axis, which limits the full utilization of computing resources and is not conducive to further improving the model's inference efficiency.
3. Formatting issue: There is a formatting issue on page 3 of the paper, with two figures each occupying a whole page. In terms of content and layout, this arrangement may not be necessary, affecting the compactness of the page and the coherence of reading.

---

> ### Author Response · Authors · 2025-03-29
> **Response to Reviewer MzYT**
>
> Thank you for your valuable review comments. Below are our responses to the issues you raised:
>
> **Comment 1: Small organ segmentation effect issue: Although the overall segmentation performance is good, there are still challenges in segmenting small organs such as adrenal glands and duodenum, with unclear boundary segmentation, which may affect the accurate diagnosis of related diseases.**
>
> Thank you for pointing out this issue, As for the problem of small organ segmentation effect issue, we will discuss it in depth in the subsequent work.
>
> **Comment 2: ROI localization limitation issue: Only the Z-axis ROI is determined, without considering the ROI localization of the X-axis and Y-axis, which limits the full utilization of computing resources and is not conducive to further improving the model's inference efficiency.**
>   Thank you for pointing out this issue, we will discuss it in depth in the subsequent work.
>
> **Comment 3: Formatting issue: There is a formatting issue on page 3 of the paper, with two figures each occupying a whole page. In terms of content and layout, this arrangement may not be necessary, affecting the compactness of the page and the coherence of reading.**
>
>   Thank you very much for pointing out the formatting issues. We have made the adjustments in accordance with your suggestions.

---

### Official Review · Reviewer_TwSi · 2025-02-17
**Review of "Locate, Crop and Segment: Efficient abdominal CT image segmentation on CPU"**

**Rating:** 8
**Confidence:** 4

**Review:**

The article "Locate, Crop and Segment: Efficient abdominal CT image segmentation on CPU" focuses on tackling the challenge of achieving efficient abdominal CT image segmentation on CPU devices. The research revolves around Task 2 of the MICCAI FLARE 2024 challenge, specifically employing a multi-scale knowledge distillation method to train the student model, thereby preserving the segmentation performance of the teacher model. To better address the segmentation issues of small organs and to resolve the quality issues of pseudo-labels, the article proposes the design of a weighted composite loss function. Additionally, a Z-axis region of interest (RoI) localization strategy based on the liver is proposed to accelerate inference speed.

Issues:
1. Insufficient analysis of the impact of pseudo-label quality. The paper mentions using pseudo-labels from FLARE 2022 to train the student model but does not quantitatively analyze the specific impact of pseudo-label noise on model performance (e.g., through ablation experiments comparing the effects with and without pseudo-labels).
2. Insufficient depth in result discussion. The analysis of segmentation failure cases (e.g., pancreas, inferior vena cava) only stays at anatomical explanations, without in-depth exploration of the limitations of the model structure or training strategies. It is recommended to analyze the reasons from the perspectives of feature learning, loss function design, etc., and propose targeted improvement plans.

---

> ### Author Response · Authors · 2025-03-29
> **Response to Reviewer TwSi**
>
> We sincerely thank Reviewer TwSi for the valuable comments and constructive feedback. Below are our point-by-point responses addressing the concerns raised.
>
>  **Comment 1: Insufficient analysis of the impact of pseudo-label quality. The paper mentions using pseudo-labels from FLARE 2022 to train the student model but does not quantitatively analyze the specific impact of pseudo-label noise on model performance (e.g., through ablation experiments comparing the effects with and without pseudo-labels).**
>
> Thank you very much for your valuable feedback. We truly appreciate your interest and input. In the FLARE22, there were indeed participants who conducted research on the impact of pseudo-label noise on model performance and compared the effects with and without pseudo-labeling. Since this topic has already been explored by others in the same context, we have chosen not to cover it again in our current work to avoid redundancy. We hope you understand our decision and thank you once again for your understanding and support.
>
> **Comment 2: Insufficient depth in result discussion. The analysis of segmentation failure cases (e.g., pancreas, inferior vena cava) only stays at anatomical explanations, without in-depth exploration of the limitations of the model structure or training strategies. It is recommended to analyze the reasons from the perspectives of feature learning, loss function design, etc., and propose targeted improvement plans.**
>
> Thank you for your valuable advice. We have deepened the discussion by extending the analysis of segmentation failure cases to the limitations of model architecture and training strategies, and have proposed targeted improvement plans.

---

### Official Review · Reviewer_FUKh · 2025-03-11
**Typos and style**

**Rating:** 9
**Confidence:** 5

**Review:**

Typos and style: There are many typos or formatting issues, such as:

“...the segmentation accuracy.In order to...” missing space;

“...biomarker measurement systems[27]. Segmentation...” missing space;

“...computational overheads.(2) In order to improve...” missing space;

“... ROI range is (i + minzidx- 7/8 * patch_size[0], i + minzidx + 2 * 7/8 * patch_size[0]).” change '*' to '×';

“...based on running time, as shown in Table 7” missing punctuation;

Table 6 is missing the bottom border line.

---

> ### Author Response · Authors · 2025-03-29
> **Response to Reviewer FUKh**
>
> Thanks for the suggestion, the misspelling or formatting issues in the article have been fixed.

---

### Decision · Program_Chairs · 2025-03-20

**Decision:**

Accept

**Comment:**

Please carefully address the reviewers' comments in the revision.

---

> ### Comment · Program_Chairs · 2025-03-31
>
> Sec 4.4 is not completed.
>
> Point-to-point response is not available.

---

> > ### Author Response · Authors · 2025-04-01
> > **Response to Program Chairs**
> >
> > Thank you for your feedback. We have carefully addressed all the reviewers' comments and revised the manuscript accordingly. A detailed response to each point is included in the revised submission, and the requested test results have been incorporated into the manuscript.

---

> ### Author Response · Authors · 2025-04-01
> **Response to Program Chairs**
>
> Thank you for your feedback. We have carefully addressed all the reviewers' comments and revised the manuscript accordingly. A detailed response to each point is included in the revised submission, and the requested test results have been incorporated into the manuscript.